# Thrombotic Events during Lenvatinib Treatment: A Single Institution Experience

**DOI:** 10.3390/jcm11247312

**Published:** 2022-12-09

**Authors:** Nerina Denaro, Ornella Garrone, Michele Ghidini, Gianluca Tomasello, Jens Claus Hahne, Marco Carlo Merlano, Laura Deborah Locati

**Affiliations:** 1Oncologia Medica Fondazione IRCCS Ca’ Granda Ospedale Maggiore Policlinico, 20122 Milan, Italy; 2Division of Molecular Pathology, The Institute of Cancer Research, London SM2 5NG, UK; 3Scientific Direction Candiolo Cancer Institute FPO-IRCCS, 10060 Candiolo, Italy; 4Translational Oncology, IRCCS ICS Maugeri, 27100 Pavia, Italy; 5Department of Internal Medicine and Therapeutics, University of Pavia, 27100 Pavia, Italy

**Keywords:** lenvatinib, cerebrovascular events, radio-refractory differentiated thyroid cancer

## Abstract

Lenvatinib is the standard treatment for radioiodine-refractory differentiated thyroid cancer (RR-DTC). Thromboembolic (TE) side effects are quite rare (1–3% of treated patients) in clinical trials. Nevertheless, patients with predisposing factors are at a higher risk of developing cardiovascular adverse events. Reduction of lenvatinib starting dose and cardiologic counselling to provide appropriate supportive therapies are usually recommended for high-risk patients. From 2016 to 2022, we analyzed a series of 16 patients who were consecutively treated at our institution. All except one patient received a reduction in their dosage after two cycles of therapy because of toxicities, and four patients (25%) suffered from TE. The observed incidence in our patient sample seemed to be higher than expected. We hypothesized that our patient sample might be at higher risk probably because of the heavy prior loco-regional treatments performed.

## 1. Introduction

Thyroid cancer is the most prevalent endocrine tumour, affecting around 45,311 new cases/year in US (including microcarcinomas) and 13,000 new cases/year in Italy [1,2]. Differentiated thyroid cancer (DTC) can be treated with surgery and appropriate radioiodine (RAI), with a global overall survival rate of 90% at 10 years, but some patients may recur after surgery. Radioiodine is accepted as the first-line therapy for recurrent/metastatic DTC patients. Unfortunately, about 50% of these patients develop radioiodine resistance at some points during their clinical history. This is more common in the presence of one of the following criteria: no RAI uptake since diagnosis or during treatment, disease progression within 12 months despite RAI avidity, and extensive RAI exposure (>600 mCi). Frequently, these criteria overlap.

In patients with high tumor burden and rapid progressive or symptomatic radioiodine-refractory differentiated thyroid cancer (RR-DTC), systemic therapy is needed.

The treatment of RR-DTC has changed with the advent of tyrosine kinase inhibitors (TKIs) [3]. Lenvatinib (Lenvima^®^) is a potent TKI that selectively inhibits the kinase activities of vascular endothelial growth factor receptors VEGFR1 (also known as FLT1), VEGFR2 (KDR), and VEGFR3 (FLT4). In addition, it inhibits other pro-angiogenic and oncogenic pathways related to receptor tyrosine kinases (RTKs), including all classes of fibroblast growth factor receptors (FGFR 1-4), platelet-derived growth factor receptor α (PDGFRα), KIT, and RET [3]. Lenvatinib is currently approved as a monotherapy for the treatment of adult patients with progressive, locally advanced, or metastatic RR-DTC (papillary/follicular/Hürthle cell) and for the treatment of patients with advanced or unresectable hepatocellular carcinoma who have received no prior systemic therapy. In addition, it is recommended as a first-line treatment for recurrent/metastatic adenoid cystic carcinoma (ACC) of salivary glands in recent ASCO guidelines [4]. Lenvatinib in combination with pembrolizumab has been approved for the treatment of adult patients with advanced or recurrent endometrial carcinoma who have progressed on or following prior treatment with platinum-containing therapy in any setting and who are not a candidate for curative surgery or radiation. Even if cardiovascular complications are the most common adverse events correlated with lenvatinib therapy, the rate of TE is uncommon both in clinical trials (3%) and in real-world experiences, with only one study reporting incidence over 20% (range 3–21%), regardless of the tumour type [5,6,7,8,9,10]. Indeed, Koehler et al. reported data similar to our small case series (serious complications, e.g., haemorrhage, acute coronary syndrome, and thrombosis/venous thromboembolism in 21% patients taking lenvatinib), while Platini F et al. reported a rate of 0.6% of TE and other authors did not report a high incidence of thromboembolic events [5,6,7,8,9,10]. Cerebral stroke is one of the most serious complications of TE. The general risk factors for stroke include hypertension, heart disease (comprising valvular disease and arrhythmia), diabetes, smoking, oral contraceptives, history of transient ischemic attacks, high red blood cell count, high blood cholesterol and lipids, obesity, lack of exercise, excessive alcohol use, older age, history of prior stroke, and hereditary factors or genetics [11,12,13]. Moreover, previous radiotherapy involving supra-aortic vessels or surgery in the neck area are additional risk factors for TE [11,12,13]. Unfortunately, several experiences coming from low-volume centres have demonstrated a higher number of futile surgeries on the neck and thyroid beds. In the pivotal phase III of the SELECT trial, arterial TE were reported in 3% of RR-DTC patients treated with lenvatinib compared to 1% in the placebo group. Most of the affected patients had some predisposing factors, such as cardiovascular disease (hypertension, atrial fibrillation, and valvular diseases) and obesity [3]. In a recent study based on 32 patients with adenoid cystic carcinoma (ACC) treated with lenvatinib, 15.6% of the patients had partial response (PR) and 75% of the patients had stable disease (SD) [14]. Around 28% of the patients experienced grade 3 hypertension, 9.4% grade 3 oral pain, 0.3% intracranial haemorrhage, and 0.3% acute coronary syndrome. No grade 5 toxicity was recorded [14]. However, these data on TE were not confirmed in an Italian phase II trial enrolling 26 ACC patients. Grade 3 toxicity was observed in 50% of the patients (25% asthenia, 21% hypertension, 4% decreased weight, and 4% stomatitis), and neither grades 4 and 5 nor TE side events were recorded [15]. Platini et al. described the burden of late side effects in a series of 37 RR-DTC patients treated with lenvatinib at the Istituto Nazionale dei Tumori (Milan, Italy). Late side effects were defined as new side effects of any grade that appeared after 12 months of lenvatinib therapy. In this study, cardiovascular toxicity was the most common late toxicity (57%), with a rate of 0.6% for TE events [16].

## 2. Materials and Methods

We reported 4 cerebrovascular events in 16 patients who were consecutively treated with lenvatinib between 2016 and 2022. All patients were treated with the standard dosage of lenvatinib, 15 patients experienced RR-DTC and one suffered from inoperable adenoid cystic carcinoma (ACC). Lenvatinib at a starting dose of 24 mg daily was administered. The first dose reduction was 20 mg, the second dose reduction was 14 mg, and the third dose reduction was 10 mg.

The patients’ characteristics are summarised in Table 1. The characteristics of the patients who suffered from cerebrovascular events are shown in Table 2, and the analytical results from the blood samples of these patients are summarised in Table 3.

## 3. Results

### 3.1. Case 1

Case 1 was a female patient, aged 72 years old at the time of the cerebrovascular event in November 2020, who had been on lenvatinib therapy since 2015. She was a current smoker and smoked 5 cigarettes per day, with a smoking rate of 13 pack per year in the past. No significant medical events were reported in the past. Her performance status (PS) was 0. In January 2005, she underwent total thyroidectomy for follicular carcinoma (pT3, Nx, Mx). She received postoperative (radioactive iodine) RAI ablation therapy. In November 2005, she underwent neck dissection (ND) due to nodal recurrence (5/43 metastatic nodes of follicular thyroid disease) and, subsequent, underwent RAI in April and December 2006, July 2007, February 2009, and May 2011 (cumulative dose >600 mCi). In April 2015, a brain progression at the right cerebellum was documented. The patient underwent excision of residual thyroid, and the pathology report confirmed the diagnosis of follicular carcinoma. There was no neurosurgical indication for brain metastasis. The patient started lenvatinib at 24 mg daily in July 2015. Her ECOG PS was 0, baseline thyroglobulin (TG) was >3000 ng/mL, and antibody (Ab) anti-TG level was normal. The patient achieved excellent biochemical response in two months (TG < 20 ng/mL). Adverse events were reported according to CTCAEv4. Treatment-related side effects were mild with grade 2 fatigue, grade 2 hypertension (despite the introduction of adequate anti-hypertension therapy with nebivolol, lercarnidipine, and enalapril), and grade 2 weight loss. Figure 1 shows the response on lung metastases. From the 4th cycle, lenvatinib dose was progressively reduced to 20 mg daily and then to 14 and 10 mg due to palmar-plantar dysesthesia syndrome (grade 3) and body weight reduction (grade 2). After one year of treatment, the patient refused to continue therapy due to the persistence of moderate toxicity and, finally, lenvatinib was suspended in June 2016. In November 2016, she received stereotactic RT for disease progression in a single lung node. The patient restarted lenvatinib therapy in December 2016 at a dose of 10 mg per day with overall good tolerance (with the benefits from mouth wash for grade 2 mucositis and 20% urea-based cream for grade 1 hand and foot syndrome). She obtained disease stabilization at follow-up imaging reassessments (April 2017 and, subsequently, every four months). In November 2020, she suffered from SARS-CoV-2 infection and was treated at home with low-molecular-weight heparin and steroids, while maintaining lenvatinib at 10 mg daily from Monday to Friday (she remained COVID-19 positive for 17 days). On 30 November, she was admitted to the emergency ward for ischemic stroke in the left middle cerebral area with intracranial artery occlusion and ipsilateral cerebral media artery thrombosis. She recovered with permanent impairment due to paraparesis. The patient is still alive with stable disease.

### 3.2. Case 2

Case 2 was a male patient aged 75 years old at the time of the cerebrovascular event in January 2021. His ECOG PS was 0. There was no previous history of thyroid disease nor allergy. The patient was a former smoker (30 packs per year until 20 years ago). He reported cardiovascular disease in the past: he had replacement of the aortic valve with biological prosthesis in 2013, and he suffered from hypertension and atrial fibrillation while under treatment.

In March 2014, he underwent total thyroidectomy and ND for papillary carcinoma pT3 and pN1b (10/12), followed by postoperative RAI in May 2015. In November 2015 and February 2016, he experienced neck nodal recurrence and was treated with ND and RAI (150 mCi).

In December 2017, the patient underwent the third RAI therapy for further nodal recurrence (150 mCi), with a total dose of 450 mCi. In March 2019, RR disease was reported because there was no RAI avidity. Re-staging with 18-FDG PET/CT scan identified bone, lung, and mediastinal metastatic nodes, as well as brain metastases, with pathologic growth from the cranial theca through the brain. The patient started therapy with lenvatinib at 24 mg daily in March 2020, with baseline Tg > 3000 ng/mL. A pre-treatment cardiovascular check-up was performed due to the clinical history of the patient and the higher risk of developing a cardiovascular toxicity. After two cycles of treatment, lenvatinib dosage was reduced to 20 mg daily due to grade 2 persistent dysphagia and dysphonia. The patient was followed-up for the known atrial fibrillation, which is a well-established risk factor for TE event. Further dose reduction to 14 mg per day was prescribed after four cycles due to grade 2 weight loss.

The patient achieved a major response, as shown in Figure 2.

From October 2020 onwards, lenvatinib dosage was reduced to 10 mg per day due to persistent grade 2 gastro-enteric toxicity (dysphagia and weight loss), and Tg level decreased rapidly to 20 ng/mL. On 2 January 2021, he was admitted to the emergency room for ischemic stroke. The stroke involved the middle cerebral artery and was not located in the area of brain metastasis. He died during the life support procedures.

### 3.3. Case 3

Case 3 was a female patient aged 70 years old at the time of the cerebrovascular events in March 2021. She was a former smoker (40 packs per year). There was no significant medical history until April 2013. Her PS was 2 (up and about more than 50% of waking hours). For TIR5 thyroid node, she underwent a radical thyroidectomy plus left ND for follicular thyroid cancer staged as pT3, pN1 (2/6), and Mx in May 2013. She received RAI ablation therapy. In June 2015, a follow-up CT scan revealed brain metastases (two masses that were two centimetres in the major diameter and adhered to the cranial theca). After a multidisciplinary team evaluation in October 2015, cerebral metastases were resected with the diagnosis of follicular thyroid carcinoma. The radiation oncologist did not indicate postoperative external beam whole brain radiotherapy. The patient received subsequent RAI treatment (150 mCi). In March 2016, recurrence of left neck nodes was reported and the follow-up ND confirmed 3/6 positive nodes. Unfortunately, the patient developed multiple bone metastases requiring further RAI in October 2016, January 2018, and April 2019 (150 mCi each time), with a cumulative dose of 750 mCi. In February 2021, due to the worsening of cognitive function, the endocrinologist requested imaging reassessment. A brain magnetic resonance imaging (MRI) demonstrated osteolytic lesions of the cranial theca and leptomeningeal infiltration. On 7 February 2021, she started lenvatinib at a dose of 20 mg daily. She was considered a high-risk patient for performance status due to high tumour burden and frequent hospitalization as a result of neurologic impairment; therefore, she underwent clinical examination every week. On 17 March 2021, she was admitted to the emergency department because of confusion, apraxia, and dysarthria. A brain computed tomography (CT) scan indicated haemorrhage in the left frontal area. The haemorrhagic region was not involved by considerable leptomeningeal infiltration. At the time of her visit to the emergency department, the patient was drowsy, did not reawaken to verbal stimulation, and had no finalistic response to nociceptive stimulus of the plantar in bilateral extension. She was admitted to the neurology ward and died of acute haemorrhagic stroke four days later.

Her basal CT scan with ischaemic signs is shown in Figure 3.

### 3.4. Case 4

Case 4 was a male patients aged 56 years old at the time of the cerebrovascular event in November 2021. He had neither family history of thyroid disease nor allergies. He was a current smoker (20 packs/year). In December 2014, he was diagnosed with ACC of the right hard palate. He underwent multimodal treatments in the oncology ward (maxillectomy with myocutaneous flap reconstruction followed by proton radiotherapy, complicated by osteonecrosis, mandibular abscessualization, and paralysis of the VII right cranial nerve with two salvage surgeries). Feeding was by percutaneous gastrostomy from December 2016 to February 2017. In 2018, pulmonary lobectomy was performed due to progressive disease (PD). A diagnosis of lung metastasis of ACC was confirmed. In November 2018, he underwent a salvage surgery for myocutaneous flap infection. He received prolonged antibiotic therapies and hyperbaric treatment. The patient reported surgical outcomes with distortion of the appearance of the face. In January 2019, progression of disease at the right neck nodes (Robbins levels II and III) was confirmed by CT scan and 18-FDG PET/CT scan. Platinum-based chemotherapy (doublet) started in June 2019 with major response. His disease remained stable, as shown on CT and 18-FDG PET/CT scans, up until August 2021. A head and neck MRI performed in September 2021 did not record vessel alterations. However, disease progression was documented with “Solid tissue at the internal posterior corner of left maxilla with bone lysis and infiltration along the posterior sinus wall”. He started lenvatinib at 24 mg daily in September 2021. His metabolic response was recorded using 18-FDG PET/CT scan in November 2021. Persistence of uptake at the tongue-oropharynx residue and hyoid bone, as well as a reduction to the maxillary sinus extended to the ethmoidal cells, were described; there were no other known metastatic sites. The MRI showed a reduction of neoplastic tissue. On 30 November 2021, the patient was admitted to the emergency room for left hemiplegia, bradypnea, desaturation, and hypotension. Neurologic examination described the patient as being sleepy and unresponsive, with response only to nociceptive, indifferent plantar cutaneous reflex, medium-intensity osteotendinous reflex not being clearly prevalent, and slight anisocoric pupils (left > right, Glasgow Coma Scale 4). A CT scan showed alterations in the parenchymal density of the right occipital-parietal area with gray-white densitometry gradient attenuation and overall ASPECT score of about 7. A CT angiography scan documented occlusion of the right common carotid at the origin, including the internal and external branches to the intracranial siphon and regular opacification of the right middle cerebral artery originating from contra-lateral compensation (Figure 4); a lack of opacification of the right M3 distal arterial branch in place was recognized. No indication for endovascular treatment was made. On the neurological side, considering the extension of the ischemic core (>2/3 right medium cerebral arteria) and the oncological therapy in progress (which increased both thromboembolic and hemorrhagic risks), there was no indication for intravenous thrombolytic treatment. An extensive right basal thickening was recorded in the lung. Despite supportive care, the patient died within one month.

## 4. Discussion

In this limited series of cases, we recorded four vascular events (three fatal) that seemed to exceed the frequency expected on the basis of the published major studies (25% vs. 3–5%) [3,4,5,6,7,8,9,10,11,12,13,14]. Several reasons, such as age, smoking status, and predisposing factors, could help explain this difference. For example, the median age of our series of 16 patients was slightly higher than in the published studies (median age 68.5 years vs. 64 years in the SELECT study), and three patients were older than 70 years All our patients were active smokers, while the SELECT study and other real-world experiences did not report information about smoking status, a well-known predisposing factor [10,15,16]).

In our study sample, vascular events occurred in the first six months in two patients and after nearly two years of treatment in the other two patients. Previous number of surgical therapies might correlate with this high risk: our patients underwent thyroidectomy or tumour excision and at least two neck dissections. Moreover, one patient received external radiotherapy up to 70 Gy to the neck. The younger ACC patient developed a cerebrovascular TE on the same side as his previous loco-regional treatment (surgery and radiotherapy) for the treatment of the primary tumor, and this aspect might have influenced the outcome. It must be stressed that these events happened during the COVID-19 pandemic and one patient (late side effects) recovered from a COVID-19 infection a few days earlier. This might be an additional risk factor in this patient. In a meta-analysis of 102 studies, the frequency of COVID-19-related venous thromboembolic event (VTE) and arterial thromboembolic event (ATE) were 14.7% and 3.9%, respectively [17]. One patient in our case series had COVID-19, so this most probably have contributed to the TE event. The most common toxicities observed in our study cohort were similar to the ones reported for the SELECT study [3]. Nevertheless, we observed higher toxicities, such as hypertension, decreased appetite, and weight loss, compared to the Italian Expanded Access study [10]), and this might be most probably due to the limited number of treated cases, the relatively minor expertise of our centre, and a multidisciplinary approach that has been built only in recent years (Table 4).

Hypertension has been recognized as a common risk factor for lenvatinib. High blood pressure was observed in all four cases reported with cerebrovascular events and no cerebrovascular events were recorded in the seven patients that did not have high blood pressure. An adequate control of blood pressure is recommended in order to reduce the endothelial damage.

Treatment compliance is a challenge in thyroid cancer patients, and, therefore, a lower dose of this drug has been evaluated. A study investigating lenvatinib reduction (18 mg) in thyroid cancer failed to show non-inferiority compared to the standard dose (24 mg). No difference in safety reports was documented [18]. However, in daily practice in high-risk patients (tumour with major invasion of the great vessels, trachea and oesophagus, untreated brain metastases, unhealed wounds, risk factors for intestinal perforation, cardio-vascular comorbidities, and low body mass index), a starting dose of 14 mg is generally suggested. In a large meta-analysis with more than 9000 patients treated with anti-angiogenic TKI, the risk of developing arterial events increased significantly (OR 2.6; *p* = 0.001). In this study, the most common events for TE were cardiac ischemia/infarction (67.4%), central nervous system ischemia (7.9%), and cerebrovascular accident (6.7%) [11]. In the SELECT trial dealing with RAI-refractory DTC, dose reduction and discontinuation were the common actions taken for an adverse reaction in 63.1% and 19.5% of the patients, respectively. Adverse reactions that most often led to lenvatinib discontinuation were proteinuria, asthenia, hypertension, cerebrovascular events, diarrhoea, and pulmonary embolism [3,12]. By analyzing the data from 1781 patients from 12 trials, including 4 randomized controlled trials and 8 phase II single arm trials, it was shown that the overall incidence of high-grade arterial thromboembolism events and venous thromboembolism events associated with TKIs were 1.4% and 3.3%, respectively [13]. TKI treatment significantly increased the risk of developing high-grade ATE (*p* = 0.029), but not high-grade venous thrombosis (*p* = 0.54), when compared to placebo. Cardiac ischemia (28.6%) followed by cerebrovascular events (21.4%) were the most common arterial thrombosis events [13]. In another meta-analysis involving 4679 patients treated with sorafenib, sunitinib, or pazopanib, the results demonstrated an increased risk of fatal adverse events (RR 2.23), regardless of tumour type and drug used, when compared to the control groups. Myocardial infarction represented the main cause of all deaths attributable to VEGFR inhibitor in 15% of the patients [19]. In a Japanese study of severe adverse events from lenvatinib treatment, embolism was observed in 4 out of 79 patients with DTC (pulmonary embolism, cerebral infarction, thrombotic cholecystitis, and deep vein thrombosis) and 2 out of 32 patients with anaplastic thyroid cancer (cerebral infarction and thrombotic cholecystitis) [20]. In a Korean real-world experience of lenvatinib, cerebrovascular accidents were observed in 4.1% of the patients [8]. Albeit in a smaller series, these data are in line with the studies published by Bai et al. and Kim SY [8]. Therefore, several cancer and patient related factors, in addition to antiangiogenic TKI activity, might be responsible for the higher risk of vascular events. Firstly, the timing of diagnosis of adverse events and an appropriate management are fundamental to avoid interruptions and unnecessary reductions. As Brose et al. stressed in their randomized non-inferiority study [18], it is essential to consider the characteristics of the patients (comorbidities, caregiver, and compliance to treatment) and the characteristics of the tumour (site and size of metastases). It is important to know when to stop, reduce the dose, or stop drug administration when needed. Adequate and timely treatment of collateral events and assessment of quality of life and overall response are essential [16]. After these severe TE events, we proposed a pre-treatment neurological and dietary evaluation for patients who are candidates for lenvatinib treatment. Apart from the standard examination (blood cell count; liver, kidney, and thyroid function; and glucose, cholesterol, and lipid assessment), we also evaluated erythrocyte sedimentation rate (ESR), C-reactive protein (CRP) D-Dimer test, serum protein electrophoresis, coagulation tests, and homocysteine test. Additionally, carotid ultrasound and echocardiogram were routinely performed.

## 5. Conclusions

Although a reduced starting dose of lenvatinib is not approved in thyroid and adenoid cystic cancer patients, a careful selection of patients and a rigid evaluation of risk factors might be useful in everyday practice. We are aware of the limitation of our small series, and we propose a multidisciplinary assessment during each step of disease management to reduce unnecessary locoregional treatments. Although the incidence of thromboembolism in this small series is surprising, further analysis on safety is highly recommended to identify high-risk categories. Moreover, we suggest adequate education of patients at pre-treatment and while on treatment.

## Figures and Tables

**Figure 1 jcm-11-07312-f001:**
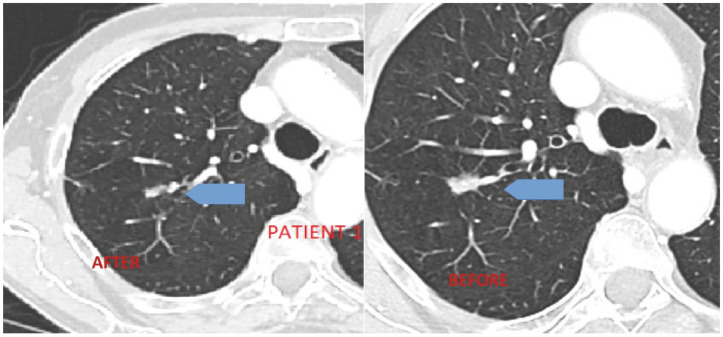
Patient 1’s response to lenvatinib treatment for lung disease.

**Figure 2 jcm-11-07312-f002:**
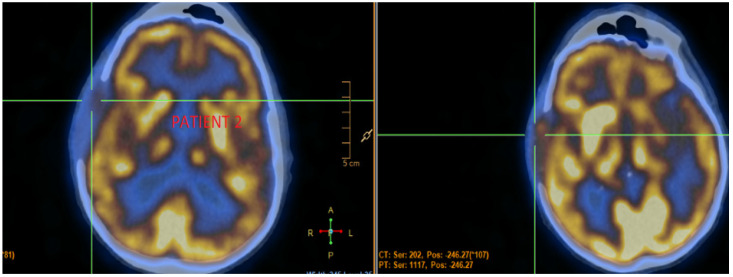
The patient’s response to lenvatinib treatment for bone disease as shown on FDG PET/CT scan.

**Figure 3 jcm-11-07312-f003:**
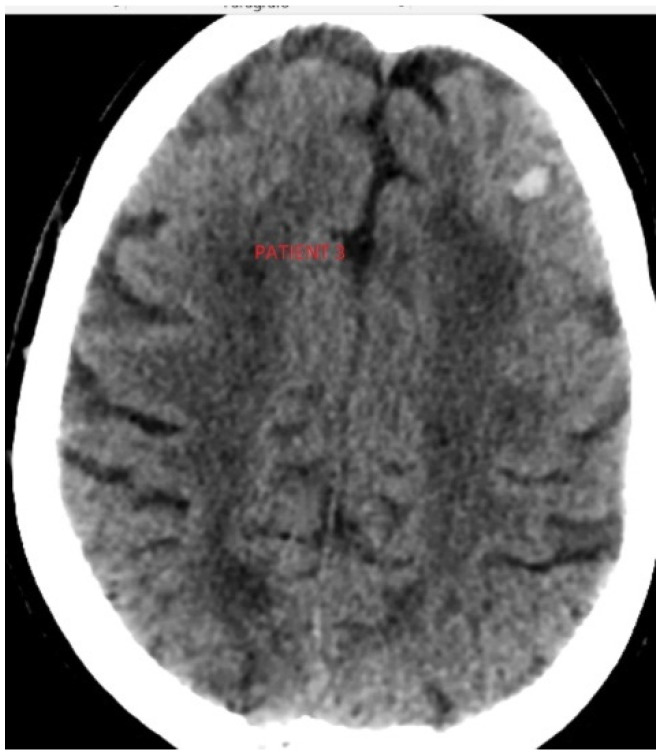
Basal CT scan after the ischemic event in patient 3.

**Figure 4 jcm-11-07312-f004:**
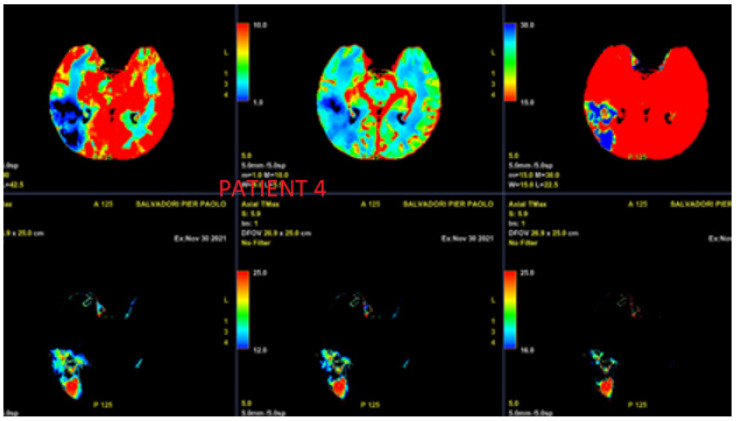
Vascular study of haemorragic disease in patient 4.

**Table 1 jcm-11-07312-t001:** Entire patient sample characteristics.

Pts	Age	Histology	Sex	Smoking History	High Cholesterol	High Blood Pressure	BMI	Start Lenv	End Lenv	TE
1 *	72	Fo	F	N	N	Y	17.2	23 July 2015	30 November 2021	30 November 2021
2	71	Pa	M	Y	Y	Y	20.6	22 May 2015	2 July 2021	
3	89	Pa	M	N	N	Y	21	20 February 2017	25 October 2017	
4	71	Pa	F	Y	N	Y	28	10 July 2017	27 December 2017	
5 *	75	Pa	M	Y	N	Y	21.9	26 March 2019	30 December 2020	30 December 2020
6	84	Pa	M	Y	N	Y	23	5 February 2018	31 December 2020	
7	54	Pa	F	Y	N	Y	17	26 April 2021	31 October 2021	
8 *	56	ACC	M	Y	N	Y	24.16	21 September 2021	30 November 2021	30 November 2021
9 *	70	Pa	F	N	N	Y	20.2	7 February 2021	17 March 2021	17 March 2021
10	65	Pa	F	Y	N	N	21	10 May 2021	ongoing	
11	59	Pa	F	N	N	N	18.9	9 August 2021	ongoing	
12	59	Pa	M	Y	N	N	22	16 August 2021	ongoing	
13	75	Pa	F	Y	Y	N	24.5	22 November 2021	ongoing	
14	40	Fo	M	Y	N	N	26.1	15 November 2021	ongoing	
15	73	Pa	F	N	Y	N	20.3	6 December 2021	ongoing	
16	78	Fo	M	Y	N	N	22.1	23 July 2015	ongoing	

Abbreviation: Pts: patients; Fo: follicular; Pa: papillary; BMI: Body mass Index; TE: thromboembolism; ACC = adenoid cistic carcinoma; F = female; M = male; Y = yes; N = No; * pts withTE.

**Table 2 jcm-11-07312-t002:** Characteristics of patients who suffered from cerebrovascular events.

	Case 1	Case 2	Case 3	Case 4
Sex	F	M	F	M
Age	72	75	70	56
Previous surgery	3	2	2	2
Lenv start	July 2015	March 2019	Feb 2021	Sep 2021
Reduction dose level	10 mg	10 mg	20 mg	14 mg
TE event	Nov 2020	Jan 2021	March 2021	Nov 2021
High cholesterol level	No	No	No	No
Diabetes	No	No	No	No
Smoking	Yes	Yes	Yes	Yes
Atrial fibrillation	No	No	No	No
Overweight	No	No	No	No
Sedentary	No	No	No	No
High blood pressure	Yes	Yes	Yes	Yes
ECOG PS 0-1	Yes	Yes	No	Yes

Abbreviation: F = female; M = male; PS: perfomarnce status.

**Table 3 jcm-11-07312-t003:** Blood values of patients who suffered from cerebrovascular events.

	Case 1	Case 2	Case 3	Case 4
WBC baseline	6.317 × 10^9^/L	8.560 × 10^9^/L	5.600 × 10^9^/L	6.520 × 10^9^/L
WBC at the event	8.15 × 10^9^/L	1.31 × 10^9^/L	6.8 × 10^9^/L	6.85 × 10^9^/L
RBC baseline	5.4 × 10^12^/L	4.5 × 10^12^/L	3.52 × 10^12^/L	4.2 × 10^12^/L
RBC at the event	5.5 × 10^12^/L	3.8 × 10^12^/L	4.1 × 10^12^/L	4.3 × 10^12^/L
PLT baseline	229 × 10^9^/L	225 × 10^9^/L	315 × 10^9^/L	258 × 10^9^/L
PLT at the event	214 × 10^9^/L	187 × 10^9^/L	250 × 10^9^/L	195 × 10^9^/L

Abbreviation: WBC: white blood cell; RBC: red blood cell; PLT: platelets.

**Table 4 jcm-11-07312-t004:** Safety of the treatment reported in clinical trials and real-world experience.

Symptom	G1-2	G3-4	SELECT G3-4	Italian Locati 2019	Relevant Austrian Rendl 2020	Taiwan Jiang 2021	Canadian Hamidi 2022	Korea Kim 2019	Japanese Takahashi 2020
Hypertension	66	25	41.1	4.7	26	15.4–30.8	59.3	73.9	60.2
Decreased appetite	40	6.25	5.4	3.0	7	0–7.7	11.1	4.3	4.5
Decreased weight	50	6.25	9.6	4.5	7	0–7.7	14.8	4.3	4.5
Oropharyngeal pain		6.25	NA	NA	NA	NA	NA	NA	NA
Palmar-plantar dysesthesia syndrome	75	12.5	3.4	1.9	2	30.8–15.4	7.4	30.4	5.8
Arthralgia	26	0	0	NR	NR	NR	0	NR	NR
Dysphonia	36	6.25	1	NR	NR	NR	0	NR	NR
TSH changes	36	6.25	NR	NR	NR	NR	25.9		NR
Diarrhoea	56	12.5	4.8	2.8	7	NR	11.1	4.3	3.8
Fatigue	64	12.5	9.2	8.2	0	23.1–7.7	0	NR	2.3
Proteinuria					0	7.7–53.8	7.4	8.7	3.8

Abbreviation = grade; NR: not reported; NA: not assessed. All numbers are percentages.

## Data Availability

Data are available at the Hospital archive.

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
