# Peer review of "Thrombotic Events during Lenvatinib Treatment: A Single Institution Experience"

_jcm, 2022, doi:10.3390/jcm11247312_

Round 1
Reviewer 1 Report
In this communication, the authors reported observations of severe vascular events, especially cerebrovascular accidents, in a limited series of 16 RR-DTC patients on Lenvatinib, with frequency higher than expected on the basis of the published major studies (25% vs 3-5%). Although the study size is small, the observation from this report is very valuable in that previously only one study reported similar TE rate in these patients. The authors have compared their observations with the very comprehensively reviewed literature on the same topic. This report could help raise the awareness of cerebrovascular events associated with Lenvatinib in this population.
Hypertension has been recorded to be a common risk factor for Lenvatinib. High blood pressure was observed in all 4 cases reported with cerebrovascular events and no cerebrovascular events was recorded in the 7 patients that did not have high blood pressure. Hypertension is also an important risk factor for cerebrovascular events. RTI has been suggested to induce endothelial damage (PMID 33567864), which may contribute to hypertension. What is the author’s opinion on giving anti-hypertension or endothelial protective drugs (PMID: 33530139) for patients with hypertension on Lenvatinib. Did most of the patients with hypertension in this study get anti-hypertension drugs?
RR-DTC should not be abbreviated in the abstract.
Reviewer 2 Report
In this study, 15 patients with thyroid cancer and one with adenoid cystic carcinoma who received lenvatinib were studied in terms of thromboembolic events. I believe this article will be helpful in terms of making the readers more familiar with this drug and its side effects.
1- In the introduction part:
Correct repeated thyroid cancer. "Thyroid cancer is Thyroid cancer is the most..."
2- Please provide an explanation regarding the treatment protocol in the method section.
Reviewer 3 Report
I read with great interest the manuscript entitled “Thrombotic events during Lenvatinib treatment. A single institution experience.” has been evaluated. The authors analyzed a series of 16 patients consecutively treated with Lenvatinib for refractory differentiated thyroid cancer (RR-DTC) in their institution. All except one received a reduction of dose after two cycles of therapy for toxicities, and four patients (25%), suffered from thromboembolic (TE). After this severe TE, the authors proposed a pre-treatment neurological and dietary evaluation of the patient’s candidate for Lenvatinib. We evaluated, apart from standard examination (blood cell count, liver, kidney, and thyroid function, glucose, cholesterol, and lipid assessment), also erythrocyte sedimentation rate (ESR), C-reactive protein (CRP) D-Dimer test, serum protein electrophoresis, coagulation tests, and homocysteine test. Additionally, carotid ultrasound and echocardiogram were routinely performed The objective of this study is well-defined. The methodology is well written. The discussion and conclusion explain the significant outcomes of this manuscript.
The manuscript has a lot of typo mistakes, in the introduction section, the first line Thyroid cancer is repeated two times.
The table mentioned some places as “table”. It should be “Table”
There are several paragraphs throughout MS, the authors can merge them appropriately it.
Several abbreviations are used without descriptions. Definitely, abbreviations like, for instance, RR-DTC, SELECT, ECOG are commonly used. Nevertheless, corresponding descriptions would be helpful for those readers, who are not familiar with those abbreviations. Authors can introduce in the text descriptions of some abbreviations or add a list of abbreviations.
Figures 1 to 4 were not cited in the text. It should be cited appropriately.
In line 308 Bai et al. [13] reference should be added.
